# Joint Error Estimation and Calibration Method of Memory Nonlinear Mismatch for a Four-Channel 16-Bit TIADC System

**DOI:** 10.3390/s22072427

**Published:** 2022-03-22

**Authors:** Jianwei Zhang, Yuxiang Zhou, Jiaqing Zhao, Guangshan Niu, Xiangdong Luo

**Affiliations:** School of Information Science and Technology, Nantong University, Nantong 226019, China; 2010310048@stmail.ntu.edu.cn (J.Z.); 2010310040@stmail.ntu.edu.cn (Y.Z.); 2110310052@stmail.ntu.edu.cn (J.Z.); 1930310003@stmail.ntu.edu.cn (G.N.)

**Keywords:** time-interleaved analog-to-digital converter (TIADC), memory nonlinear effect, joint estimation, Volterra series, mismatch error

## Abstract

Memory nonlinear error greatly reduces the performance of analog-to-digital converters (ADCs), and this effect is more serious in a time-interleaved analog-to-digital converter (TIADC) system. In this study, the sinusoidal wave fitting method was adopted and a joint error estimation method was proposed to address the memory nonlinear mismatch problem of the current TIADC system. This method divides the nonlinear error estimation method into two steps: the nonlinear mismatch error is coarsely estimated offline using the least squares (LS) method, and then accurately estimated online using the recursive least squares (RLS) method. After the estimation, digital post-compensation method is adopted. The obtained error parameters are used to reconstruct the error and then the reconstructed error is reduced at the output. This study used a four-channel 16-bit TIADC system with an effective number of bits (ENOB) value of 10.06 bits after the introduction of a memory nonlinearity error, which was increased to 15.42 bits after calibration by the joint error estimation method. As a result, the spurious-free dynamic range (SFDR) increased by 36.22 dB. This error estimation method can improve the error estimation accuracy and reduce the hardware complexity of implementing the error estimation system using a field programmable gate array (FPGA).

## 1. Introduction

Due to the current semiconductor material and limitations of the manufacturing process, analog-to-digital converters (ADCs) are unable to maintain a high conversion accuracy at a very high conversion speed. There is an irreconcilable contradiction between conversion speed and conversion accuracy. As the only interface between analog and digital parts of electronic devices, the contradictory relationship between conversion speed and conversion accuracy of ADCs has limited the development of digitalization to a certain extent. The use of time-interleaved analog-to-digital converter (TIADC) technology can theoretically increase the conversion speed of ADCs without reducing ADC conversion accuracy.

The block diagram of the four-channel TIADC system and the trigger clock signal of each ADC are shown in Figure 1. The basic principle of the TIADC system is that *M* ADCs are interleaved sampled under the trigger of *M* interleaved clock signals. After the conversion, the converted data is output at the interleaved time by multiplexing technology, which can theoretically increase the conversion speed of the original single chip ADC by *M* times [1]. However, the TIADC system has strict matching requirements for the ADCs in the system, and a slight mismatch of the ADCs in the system will significantly reduce the performance of TIADC.

Time skew mismatch, gain mismatch, and offset mismatch of ADCs have a large impact on the TIADC system, and the technology used to calibrate the effects of these mismatches has been well studied to date [2,3,4,5,6]. The gain mismatch error and offset mismatch error are linear errors, which are relatively easy to calibrate, and their calibration method is now mature. Therefore, among the three major mismatch errors, many studies have been undertaken on the calibration of time skew mismatch errors and many calibration methods have been proposed [7,8]. In recent years, due to the improvement in calibration techniques for the mismatches mentioned above in TIADC systems, research has increasingly begun to focus on other mismatch errors for TIADC systems [9,10,11]. This paper focuses on the estimation of the memory nonlinear mismatch error of TIADC and its calibration.

A previous paper [12] proposes a static nonlinear model of TIADCs, which can model the differential nonlinearity (DNL) and integral nonlinearity (INL) of ADCs well. However, it does not consider the memory effect of the internal electronic components in ADCs. Moreover, although the calculation of its calibration is simple, it is not ideal for real systems, and its practical calibration effect is limited.

Currently, the description of memory systems or nonlinear systems is usually described by the Wiener model, Hammerstein model, Wiener-Hammerstein model, Volterra model, etc. Among these, the Volterra model is an extension of the power series and has many unique advantages in describing the memory nonlinearity of TIADC systems. It can not only describe the nonlinear mismatch in TIADC systems and the effect of memory effects on the system at the same time, it can also describe the effect of mismatches generated by gain, offset, and frequency response mismatches. Due to these advantages, the Volterra series is widely used in describing systems in electronics, remote sensing, medical, and other fields [13,14,15,16,17,18]. However, the main difficulty in modeling memory nonlinearity for TIADC systems using the Volterra model currently lies in the estimation of the kernels for each order of the Volterra series.

A previous paper [19] proposes an offline calibration method that uses a least squares (LS) method for the TIADC output to estimate the error parameters of the memory nonlinear mismatch. Its method design is simple and the error parameter estimation can achieve great accuracy when the quantity of input data is large enough, but it eventually leads to slow calculation speed and cannot meet the system requirements for real-time conversion speed. In addition, the required storage capacity for the system is quite large.

In this paper, a joint error estimation method based on the LS method for coarse estimation offline and the recursive least squares (RLS) method for accurate estimation online is proposed to address the problems mentioned above based on the Volterra series. The obtained error parameters are then used to reconstruct the error signal, and the error-free signal is obtained by subtracting the reconstructed error signal from the error-containing output signal by means of digital post-compensation.

The time complexity of the LS method is $O\left(n^{3}\right)$, but that of the RLS method is $O\left(n^{2}\right)$. Thus, the joint error estimation method proposed in this paper can greatly reduce the computational complexity compared with the traditional LS method, thus reducing the requirement of the calibration method in terms of the computational power of the calibration system. At the same time, the traditional LS method needs to store about $n^{2}$ data points but this paper only needs to store n data points, which reduces the requirement of the calibration method in terms of the storage capacity of the calibration system. It also performs fewer complex field operations when calculating the kernels of Volterra series, thus improving the efficiency of the system. After the offline error estimation, the system uses the RLS method online to approximate the more accurate error parameters. The RLS method is more suitable for the online estimation compared to the LS method because it only needs to input the current state for simple recursive operations, whereas the LS method needs to perform a large number of complex field operations after adding windows when estimating the mismatch error in the TIADC system, thus involving high computation complexity and slow computation speed.

## 2. Discrete-Time Volterra Series Model for TIADC

This section introduces the Volterra series model established for a single-chip ADC and eventually extended to the TIADC system, then the frequency domain analysis of TIADC system is carried out.

### 2.1. Discrete-Time Volterra Model for a Single-Chip ADC

In general, ADCs have a low nonlinear error, and the higher-order nonlinear error has less impact on the system [12]; thus, this paper ignores the nonlinear effects of the third and higher orders when building the Volterra model. After a certain time, for electronic devices with a memory fading effect, such as ADC, the higher-order memory effect has less impact on the system [20]. Similarly, this paper assumes that the memory order of ADCs is three, which means that the current output state of ADCs is only affected by the current input state and the previous two input states, and the previous input states have no impact on the current output state. Based on this, the discrete-time Volterra series model of a single-chip ADC is established as follows:(1)$$\begin{array}{cc} y(n) & =V[x(n)]\\ & =h_{0}+\sum_{k_{1}=0}^{2}h_{1}(k_{1})x(n-k_{1})\\ & +\sum_{k_{1}=0}^{2}\sum_{k_{2}=0}^{2}h_{2}(k_{1},k_{2})x(n-k_{1})x(n-k_{2}) \end{array}$$
where $k_{i}$ is the memory length and $h_{i}$ is the coefficient kernel of each order, and the 0th-order term of the Volterra series is a constant term that indicates the magnitude of the offset error of this ADC; the first-order term of the Volterra series indicates the magnitude of the gain error of the ADC and the effect of the past state on the current state, and the second-order term of the Volterra series indicates the memory and nonlinear effects of the ADC, i.e., the effect of memoryless nonlinearity versus memory nonlinearity on the system. The block diagram of the Volterra series model for a single ADC is shown in Figure 2, where $H_{0}[x(n)]$ indicates the orders mentioned above.

### 2.2. Discrete-Time Volterra Series Model for a Four-Channel TIADC

The four-channel TIADC system contains four ADCs that sample the input analog signal alternately under the trigger of the interleaved time signal; thus, four interleaved infinite unit sampling sequences can be used as the model for the trigger signal when modeling the clock of the four-channel TIADC system. The sampling time interval of two adjacent ADC channels is $T_{s}$, so the expression of the unit sampling sequence of the mth ADC is:(2)$$S_{m}(n)=\sum_{k=-\infty }^{\infty }\delta (n-mT_{S}-4kT_{S})$$

When modeling a four-channel TIADC, it should be noted that the time interval between two adjacent samples of the same ADC is 4$T_{s}$, and the discrete-time Volterra series expression for the four-channel TIADC can be derived as:(3)$$\begin{array}{cc} y\left(n\right) & =\sum_{m=1}^{4}V_{m}\left[x\left(n\right)\right]*S_{m}(n)\\ & =\sum_{m=1}^{4}\left\{\begin{array}{l} \left[\begin{array}{l} h_{m,0}+\sum_{k_{1}=0}^{2}h_{m,1}(k_{1})x(4n-k_{1})\\ +\sum_{k_{1}=0}^{2}\sum_{k_{2}=0}^{2}h_{m,2}(k_{1},k_{2})x(4n-k_{1})x(4n-k_{2}) \end{array}\right]\\ {}*\sum_{k=-\infty }^{\infty }\delta (n-mT_{S}-4kT_{S}) \end{array}\right\} \end{array}$$

The TIADC system block diagram is shown in Figure 3.

For computer programs, infinite sequences cannot be realized; thus, for infinite unit sampling sequences, the Poisson summation formula can be used to convert infinite sequences to the finite complex number for computation. The Poisson summation formula is as follows: (4)$$\sum_{n=-\infty }^{+\infty }f(t+nT_{S})=\sum_{K=-\infty }^{+\infty }\frac{1}{T_{S}}F(\frac{K}{T_{S}})e^{2\pi i\frac{K}{T_{S}}t}$$
where $F\left(t\right)$ denotes the Fourier transform of $f\left(t\right)$, so that the unit sampling sequence can also be expressed as:(5)$$\begin{array}{cc} S_{m}(n) & =\sum_{k=-\infty }^{+\infty }\delta [nT_{S}-mT_{S}-4kT_{S}]\\ & =\frac{1}{4T_{S}}\sum_{k=0}^{3}e^{j\pi k(n-m)/2} \end{array}$$

Then the unit sampling sequence becomes a finite sequence. However, due to the introduction of complex numbers into the calculation, the computational complexity in the field programmable gate array (FPGA) used to calibrate the system is very high. Substituting Equation (5) into Equation (3), the TIADC expression becomes:(6)$$\begin{array}{cc} y(n) & =\sum_{m=1}^{4}V_{m}[x(n)]\;*S_{m}(n)\\ & =\sum_{m=1}^{4}\left\{\begin{array}{l} \left[\begin{array}{l} h_{m,0}+\sum_{k_{1}=0}^{2}h_{m,1}(k_{1})x(4n-k_{1})\\ +\sum_{k_{1}=0}^{2}\sum_{k_{2}=0}^{2}h_{m,2}(k_{1},k_{2})x(4n-k_{1})x(4n-k_{2}) \end{array}\right]\\ {}*\frac{1}{4T_{S}}\sum_{k=0}^{3}e^{j\pi k(n-m)/2} \end{array}\right\} \end{array}$$

### 2.3. Frequency Domain Analysis of the Discrete-Time Volterra Series Model for a Four-Channel TIADC 

In this sub-section, the discrete-time Volterra series model of the four-channel TIADC with nonlinear order 2 and memory order 3 proposed above is analyzed in the frequency domain. At the same time, these frequency-domain analyses can be easily extended to M-channel TIADC systems with nonlinear order P and memory effect K. 

The sinusoidal wave fitting method is adopted in this study. For the sake of calculation, suppose the input is a cosine signal; Fourier transform is applied to Equation (3) in Section 2.2 to obtain the frequency domain expression of the time-domain signal:(7)$$\begin{array}{c} Y\left(w\right)=\sum_{m=1}^{4}\left[F_{m,0}\left(w\right)+F_{m,1}\left(w\right)+F_{m,2}\left(w\right)\right]*S_{m}\left(w\right)\\ =\sum_{m=1}^{4}\left\{\begin{array}{c} \left\{\begin{array}{c} 2\pi h_{m,0}\delta \left(w\right)\\ +\sum_{k_{1}=0}^{2}\pi h_{m,1}\left[\delta \left(w+\Omega_{0}\right)+\delta \left(w-\Omega_{0}\right)\right]e^{-jwk_{1}T_{S}}\\ +\sum_{k_{1}=0}^{2}\sum_{k_{2}=0}^{2}\pi^{2}h_{m,1}{\left[\delta \left(w+\Omega_{0}\right)+\delta \left(w-\Omega_{0}\right)\right]}^{2}e^{-jw\left(k_{1}+k_{2}\right)T_{S}} \end{array}\right\}\\ {}*\Omega_{S}\sum_{k=-\infty }^{\infty }\delta (w-\frac{k\Omega_{S}}{4})e^{-jwT_{S}\frac{m}{M}} \end{array}\right\} \end{array}$$where
$\Omega_{S}=\frac{2\pi }{T_{S}}$, Ω_0_ is the input signal frequency and *F_m,p_*(*w*) is the Fourier transform of the *p*th Volterra series of the *m*th ADC.

In order to facilitate the understanding of the influence of error signals on the signal spectrum, the frequency domain signals are decomposed into three parts: *p* = 0, 1 and 2. The expression of the memory nonlinear mismatch error in frequency domain is:(8)$$Y_{p}(w)=\sum_{m=1}^{4}\left[F_{m,p}(w)*S_{m}(w)\right]$$

According to the property that convolution satisfies the distributive law:(9)$$Y(w)=\sum_{p=0}^{2}Y_{p}(w)$$

When *p* = 0:(10)$$\begin{array}{cc} Y_{0}(w) & =\sum_{m=1}^{4}\left[F_{m,0}(w)*S_{m}(w)\right]\\ & =\sum_{m=1}^{4}\left[2\pi h_{m,0}\delta (w)*\Omega_{S}\sum_{k=-\infty }^{\infty }\delta (w-\frac{k\Omega_{S}}{4})e^{-jwT_{S}\frac{m}{4}}\right]\\ & =\sum_{m=1}^{4}[2\pi \Omega_{S}h_{m,0}\sum_{k=-\infty }^{\infty }\delta (w-\frac{k\Omega_{S}}{4})e^{-jwT_{S}\frac{m}{4}}] \end{array}$$

It can be seen from Equation (10) that the mismatch error spectrum generated by the 0th term of the Volterra series appears at $\frac{k\Omega_{S}}{4}$ (*k* = 1, 2, 3, …).

When *p* = 1:(11)$$\begin{array}{cc} Y_{1}(w) & =\sum_{m=1}^{4}F_{m,1}(w)*S_{m}(w)\\ & =\sum_{m=1}^{4}\{\sum_{k_{1}=0}^{2}\pi h_{m,1}[\delta (w+\Omega_{0})+\delta (w-\Omega_{0})]e^{-jwk_{1}T_{S}}*\Omega_{S}\sum_{k=-\infty }^{\infty }\delta (w-\frac{k\Omega_{S}}{4})e^{-jwT_{S}\frac{m}{4}}\}\\ & =\sum_{m=1}^{4}\{\sum_{k_{1}=0}^{2}\pi \Omega_{S}h_{m,1}[\sum_{k=-\infty }^{\infty }\delta (w-\frac{k\Omega_{S}}{4}+\Omega_{0})e^{-jwT_{S}\frac{m}{4}}+\sum_{k=-\infty }^{\infty }\delta (w-\frac{k\Omega_{S}}{4}-\Omega_{0})e^{-jwT_{S}\frac{m}{4}}]e^{-jwk_{1}T_{S}}\} \end{array}$$

It can be seen from Equation (11) that the mismatch error spectrum generated by the first term of the Volterra series appears at $\frac{k\Omega_{S}}{4}\pm \Omega_{0}$ (*k* = 1, 2, 3, …).

When *p* = 2:(12)$$\begin{array}{cc} Y_{2}(w) & =\sum_{m=1}^{4}F_{m,2}(w)*S_{m}(w)\\ & =\sum_{m=1}^{4}\left\{\sum_{k_{1}=0}^{2}\sum_{k_{2}=0}^{2}\frac{\pi }{2}h_{m,2}[\delta (w+\Omega_{0})+\delta (w-\Omega_{0})]e^{-jwk_{1}T_{S}}*[\delta (w+\Omega_{0})+\delta (w-\Omega_{0})]e^{-jwk_{2}T_{S}}*\Omega_{S}\sum_{k=-\infty }^{\infty }\delta (w-\frac{k\Omega_{S}}{M})e^{-jwT_{S}\frac{m}{4}}\right\}\\ & =\sum_{m=1}^{4}\left\{\sum_{k=-\infty }^{\infty }\sum_{k_{1}=0}^{2}\sum_{k_{2}=0}^{2}\frac{\pi }{2}h_{m,2}[\delta (w+2\Omega_{0}-\frac{k\Omega_{S}}{4})e^{jw(k_{1}+k_{2})T_{S}}*+\cos(k_{2}-k_{1})T_{S}\delta (w-\frac{k\Omega_{S}}{4})+\delta (w-2\Omega_{0}-\frac{k\Omega_{S}}{4})e^{-jw(k_{1}+k_{2})T_{S}}]e^{-jwT_{S}\frac{m}{4}}\right\} \end{array}$$

It can be seen from Equation (12) that the spectrum of mismatch errors generated by the second term of the Volterra series appears at $\frac{k\Omega_{S}}{4}\pm 2\Omega_{0}$ (*k* = 1, 2, 3, …).

From the above analysis, it can be seen that the frequency spectrum produced by the high-order memory nonlinear mismatch error is greater, but its coefficient decreases with the increase in order. For the system composed of weak memory nonlinear devices such as ADCs, the influence of memory nonlinearity on the overall system decreases with the increase in order.

## 3. Mismatch Error Estimation and Calibration Method

In response to the shortcomings of the current offline LS calibration method, this paper proposes a joint error estimation method that is first coarsely estimated offline by the LS method and then accurately estimated online by the RLS method. This method reduces the amount of data storage while avoiding a large number of complex field calculations compared to other traditional methods, thus greatly reducing the demand of the calibration method on system computing power and storage capacity, which improves the efficiency of calibration and allows real-time calibration online.

### 3.1. Four-Channel TIADC Memory Nonlinear Offline LS Calibration

The use of the LS method in the estimation of mismatch errors in TIADC systems has matured, and the error parameters can be estimated by least squares fitting of the overdetermined equations consisting of the input and output signals [21]. Define the input matrix consisting of a known input signal x as:(13)$$x=\left[\begin{array}{cccccccccc} 1 & x(1) & x(5) & x(9) & x{\left(1\right)}^{2} & x(1)x(5) & x(1)x(9) & x{\left(5\right)}^{2} & x(5)x(9) & x{\left(9\right)}^{2}\\ 1 & x(2) & x(6) & x(10) & x{\left(1\right)}^{2} & x(2)x(6) & x(2)x(10) & x{\left(6\right)}^{2} & x(6)x(10) & x{\left(10\right)}^{2}\\ \vdots & \vdots & \vdots & \vdots & \vdots & \vdots & \vdots & \vdots & \vdots & \vdots \\ 1 & x(n) & x(n-4) & x(n-8) & x{\left(n\right)}^{2} & x(n)x(n-4) & x(n)x(n-8) & x{\left(n-4\right)}^{2} & x(n-4)x(n-8) & x{\left(n-8\right)}^{2} \end{array}\right]$$

Define the unit sampling sequence matrix: (14)$$C_{m}=\frac{1}{4}\left[\begin{array}{ccc} \sum_{k=0}^{3}e^{j(1-m)\pi k/2} & \cdots & 0\\ \vdots & \ddots & \vdots \\ 0 & \cdots & \sum_{k=0}^{3}e^{j(n-m)\pi k/2} \end{array}\right]$$

Define:(15)$$X=\left[C_{0}x,C_{1}x,C_{2}x,C_{3}x\right]$$

Define the 10 Volterra series kernel for the mth channel as: (16)$$\begin{array}{cccccc} h_{m}=[ & h_{m,0} & h_{m,1}(0) & h_{m,1}(1) & h_{m,1}(2) & h_{m,2}(0,0)\\ & h_{m,2}(0,1) & h_{m,2}(0,2) & h_{m,2}(1,1) & h_{m,2}(1,2) & h_{m,2}(2,2)] \end{array}$$

Define:(17)$$h={\left[h_{1}\;h_{2}\;h_{3}\;h_{4}\right]}^{T}$$

Write the output matrix as:(18)$$Y={\left[y(1)\;y(2)\;\cdots \;y(n)\right]}^{T}$$

At this point, the input-output relationship of the TIADC system can be expressed as:(19)Y=Xh

The solution of the Volterra series kernel becomes a problem of solving the coefficients of the superdeterminant Equation (19). The LS method solves the superdeterminant equation as follows:(20)$$\overset{\wedge }{h}={\left(X^{H}X\right)}^{-1}X^{H}Y$$
where $X^{H}$ is the pseudo-inverse of the matrix X, i.e., at this point the singular matrix $X^{H}$ and the singular matrix X satisfy:(21)$$\left\{\begin{array}{c} X^{H}XX^{H}=X\\ XX^{H}X=X^{H} \end{array}\right.$$

At this point, $\hat{h}$ is the least squares solution of h.

### 3.2. Four-Channel TIADC Memory Nonlinear Online RLS Estimation Method

Aiming to address the shortcomings of the current offline LS error estimation method, this paper proposes a joint error estimation method with rough estimation by the offline LS method and accurate estimation by the online RLS method. The recursive initial values used in this section are derived from the values obtained from the LS method estimation in Section 3.1. 

For ease of understanding, the nth input signal of the mth ADC of the TIADC is redefined here as $x_{m}\left(n\right)$, i.e.,
(22)$$x_{m}(n)=x(4n-4+m)$$

Define:(23)$$\begin{array}{cc} \phi_{m,n}^{T}=[ & 1\;\;\;\;x_{m}(n)\;\;\;\;x_{m}(n-1)\;x_{m}(n-2)\;\;\;\;x_{m}{\left(n\right)}^{2}\\ & x_{m}(n)x_{m}(n-1)\;x_{m}(n)x_{m}(n-2)\;x_{m}{\left(n-1\right)}^{2}\;x_{m}(n-1)x_{m}(n-2)\;x_{m}{\left(n-2\right)}^{2}] \end{array}$$
(24)$$\Phi_{m,n}=\left[\begin{array}{c} \phi_{m,1}^{T}\\ \vdots \\ \phi_{m,n}^{T} \end{array}\right]$$
(25)$$P_{m,n}={\left(\Phi_{m,n}^{T}\Phi_{m,n}\right)}^{-1}$$

The nth actual output signal of the mth ADC of the TIADC is $y_{m}\left(n\right)$; then, define:(26)$$Y_{m,n}=\left[\begin{array}{c} y_{m}(1)\\ \vdots \\ y_{m}(n) \end{array}\right]$$

Define ${\hat{\theta }}_{m,n}$ as the nth recursive estimate of the mth ADC mismatch error; then:(27)$${\overset{\wedge }{\theta }}_{m,n}=P_{m,n}\Phi_{m,n}^{T}Y_{m,n}$$

Next, based on the current input signal $x_{m}\left(n\right)$ and output signal $y_{m}\left(n\right)$, use recursion to obtain a more accurate mismatch parameter for this ADC, and define:(28)$$P_{m,n+1}=P_{m,n}-\frac{P_{m,n}\phi_{m,n+1}\phi_{m,n+1}^{T}P_{m,n}}{1+\phi_{m,n+1}^{T}P_{m,n}\phi_{m,n+1}}$$

Then, recursively obtain the next estimate as:(29)$${\overset{\wedge }{\theta }}_{m,n+1}={\overset{\wedge }{\theta }}_{m,n}+P_{m,n+1}\phi_{m,n}(y_{m,n}-\phi_{m,n}^{T}){\overset{\wedge }{\theta }}_{m,n}$$

The RLS method has many advantages, such as good adaptability to non-stationary signals, fast convergence, and high accuracy and stability of estimation. Compared with the LS method, the single computing complexity is low and the amount of stored data is small. In addition, only the current state amount needs to be input, no time window needs to be added, and it can meet the real-time requirements.

## 4. Four-Channel TIADC Online RLS Mismatch Error Estimation Compensation Structure

In this section, the calibration structure for the nonlinear mismatch error of the TIADC is presented. The sinusoidal wave fitting method is adopted in this paper. The proposed calibration method is a digital post-compensation calibration method, in which the error is reconstructed by the estimated error parameters after analyzing the signal from the TIADC output and input. Finally, the error is subtracted from the output of the TIADC to compensate the signal for the corrected signal.

As shown in Figure 4, for a given input cosine signal $x\left(t\right)$, its discrete signal $x\left(n\right)$ at any given moment n after passing through the sample and hold amplifier (SHA) is also known. $y\left(n\right)$ is the output of the TIADC system containing the mismatch error.

The input and output signals are fed to the RLS module simultaneously for recursive operations, which can be calculated as described in Section 3.2. The final RLS module outputs the computed series kernel to the error reconstruction module. Let the RLS error estimation module that performs error estimation for the mth ADC compute the output series kernel as:(30)$$\begin{array}{llllll} \theta_{m}= & [\theta_{m,0} & \theta_{m,1}(0) & \theta_{m,1}(1) & \theta_{m,1}(2) & \theta_{m,2}(0,0)\\ & \theta_{m,2}(0,1) & \theta_{m,2}(0,2) & \theta_{m,2}(1,1) & \theta_{m,2}(1,2) & \theta_{m,2}(2,2)] \end{array}$$

The function of the error reconstruction module for a single channel is:(31)$$\begin{array}{cc} E_{m}[x(n)] & =h_{m,0}+\sum_{k=0}^{2}h_{m,1}(k)x_{m}(n-k)\\ & +\sum_{k_{1}=0}^{2}\sum_{k_{2}=0}^{2}h_{m,3}(k_{1},k_{2})x_{m}(n-k_{1})x_{m}(n-k_{2}) \end{array}$$
of which:(32)$$h_{m,i}=\left\{\begin{array}{c} \theta_{m,i}-1\;,i=1\\ -\theta_{m,i}\;,\mathrm{other} \end{array}\right.$$

Then the complete error function is finally established as:(33)$$E(n)=\sum_{m=1}^{4}\left[\begin{array}{l} h_{m,0}+\sum_{k=0}^{2}h_{m,1}(k)x_{m}(n-k)\\ +\sum_{k_{1}=0}^{2}\sum_{k_{2}=0}^{2}h_{m,3}(k_{1},k_{2})x_{m}(n-k_{1})x_{m}(n-k_{2}) \end{array}\right]$$

The overall correction block diagram is shown in Figure 5, and the corrected output signal is:(34)$$y_{c}(n)=y(n)-E(n)$$

## 5. Experimental Results and Analysis

This study used SIMULINK-based simulation for verification. The input sine wave frequency was 53 MHz. The TIADC system consists of four ADCs. The simulation did not use over-sampling technology, so the conversion speed and sampling rate were the same; the overall sampling rate of the TIADC was set as 1 GHz, i.e., the sampling time interval of each ADC in TIADC system was 1 ns; the sampling rate of each internal ADC was set to 250 MHz; the resolution was set to 16 bit; and the range was ±1 V. The time-domain figure of a single ADC with a nonlinear mismatch is shown in Figure 6.

According to the Nyquist sampling theorem, in the analog-to-digital conversion process, when the sampling frequency $f_{s}$ is more than twice the highest frequency $f_{max}$ in the continuous signal input, the digital signal after sampling can restore the original signal completely. Thus, this experimental analysis intercepted a sampling rate that was only half of the spectrum, namely from 0 MHz to 500 MHz, and the other meaningless part of the spectrum was discarded.

After adding memory effect errors and nonlinear effect errors to the TIADC model, its time-domain figure is shown in Figure 7. As shown in the figure, due to the existence of mismatch error and noise, the output digital signal cannot effectively match the input continuous analog signal. Compared with a single ADC with memory nonlinear error in Figure 6, the output waveform distortion of the TIADC system with memory nonlinear mismatch is higher. Its spectrum is shown in Figure 8, where the input signal produces a peak spectrum at 53 MHz and the others are the spectra of noises generated by the mismatch error. According to the analysis in Section 2.3, the noise generated by the 0th-order memory nonlinear mismatch error is located at 0 MHz, 250 MHz and 500 MHz. The noise generated by the first-order mismatch error is located at 197 MHz, 303 MHz and 447 MHz. The noise generated by the second-order mismatch error is located at 106 MHz, 144 MHz, 356 MHz and 392 MHz. The coefficients for each of the four channels are defined as: $h_{1}=\left[47\times 10^{-5},-34\times 10^{-5},33\times 10^{-5},-26\times 10^{-5},79\times 10^{-6},-33\times 10^{-5},17\times 10^{-6},-42\times 10^{-7},-12\times 10^{-7},1\times 10^{-7}\right]$, $h_{2}=\left[43\times 10^{-5},42\times 10^{-5},-36\times 10^{-5},27\times 10^{-5},-91\times 10^{-6},-21\times 10^{-5},-28\times 10^{-6},-36\times 10^{-7},23\times 10^{-7},2\times 10^{-7}\right]$, $h_{3}=\left[33\times 10^{-5},-23\times 10^{-5},-33\times 10^{-5},-28\times 10^{-5},-77\times 10^{-6},34\times 10^{-5},39\times 10^{-6},-47\times 10^{-7},13\times 10^{-7},-9\times 10^{-7}\right]$, $h_{4}=\left[43\times 10^{-5},37\times 10^{-5},44\times 10^{-5},39\times 10^{-5},62\times 10^{-6},43\times 10^{-5},-16\times 10^{-6},58\times 10^{-7},-26\times 10^{-7},-13\times 10^{-7}\right]$.

At this point, due to the influence of memory nonlinearity, the effective number of bits (ENOB) of this TIADC with 16 bits is only 10.06 bits. Its spurious-free dynamic range (SFDR) is 64.29 dB, and its signal-to-noise ratio (SNR) is 62.35 dB.

After the first offline LS calibration, the second-order Volterra series kernel could not be effectively fitted by the LS method due to the small number of sampling points set, and divergence occurred in the fitting operation. Fortunately, since the ADC is a strongly linear system, the parameters that largely deviate from the standard linear system parameters can be easily excluded when applying the fitted parameters to the calibration structure. For the error parameters with small deviations that are not excluded, after a certain period of recursive operation, the system can also converge to the exact error parameter values.

As shown in Figure 9, there is some improvement in the calibration after excluding the divergent error parameters, and there is a significant improvement in its offset error, gain error, and first-order memory effect. According to the frequency domain analysis of the TIADC system in Section 2.3, the noise generated at 0 MHz, 197 MHz, 250 MHz, 303 MHz, 447 MHz, and 500 MHz by order 0 and 1 of Volterra series is effectively suppressed, and the noise generated by the error of order 3 of the Volterra series is also slightly improved. The SFDR becomes 76.88 dB, the SNR becomes 74.45 dB, and the ENOB becomes 12.06 bits.

After the offline LS method to obtain the error parameters, the online RLS calibration was performed with the error parameters after excluding the divergence series kernel as the recursive initial values to ensure that the recursion reaches stability. The spectrum is plotted as shown in Figure 9. It can be seen from the graph that the errors caused by various mismatches have improved significantly. At this point the SFDR improves to 100.51 dB, the SNR improves to 94.79 dB, and the ENOB improves to 15.42 bits.

After the sinusoidal fitting calibration carried out by the joint calibration algorithm proposed in this paper, to further verify the practicability of the proposed algorithm, the recursive value after stabilization was used as the error parameter to reconstruct the error to calibrate the actual signal. A random analog signal was input into the system, and the final time-domain output is partly shown in Figure 10 and the error bit comparison before and after calibration is shown in Figure 11. As is shown in the figure, compared to the output before calibration, the output digital signal after the joint calibration better fits the analog input. The comparison of calibration of different stages is shown in Table 1.

In order to verify the practicability of the proposed calibration method, analog signals of different frequencies were input into the calibration system in the experiment. Since the SNR of signals shows a linear relationship with ENOB, only the ENOB and the SFDR are plotted, and the final calibration effects of input signals with different frequencies are shown in Figure 12. Theoretically, an ADC can sample signals at frequencies less than half of the highest sampling rate according to the Nyquist sampling criterion.

## 6. Conclusions

In this study, a Volterra model of memory nonlinear mismatch of TIADC systems was established. In addition, a joint error estimation method comprising offline LS and online RLS was proposed to address various defects in the existing LS method for estimating error parameters applicable to memory nonlinear calibration. In order to verify the effectiveness of this method for memory nonlinear mismatch calibration, four ADCs with a sampling rate of 250 MHz and a resolution of 16 bits were used to form a four-channel TIADC system with a sampling rate of 1 GHz. Then, different memory nonlinear errors were added to different ADCs. Simulation experiments showed that this method can effectively fit the memory nonlinear mismatch error and ultimately be used for effective calibration to the noise caused by the memory nonlinear mismatch error of the TIADC system. This error estimation method can improve the error estimation accuracy and, at the same time, greatly reduce data storage capacity. Compared with the existing methods, it can reduce the computational complexity and storage requirements of the calibration system, and has high practicality and a low difficulty of engineering implementation. This method can also be extended to TIADC systems in which the memory effect of the ADCs increases or the linearity decreases, and to M-channel TIADCs.

## Figures and Tables

**Figure 1 sensors-22-02427-f001:**
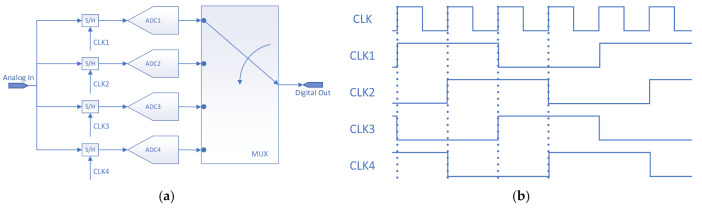
(**a**) TIADC system diagram; (**b**) time sequence diagram of each ADC in a TIADC system.

**Figure 2 sensors-22-02427-f002:**
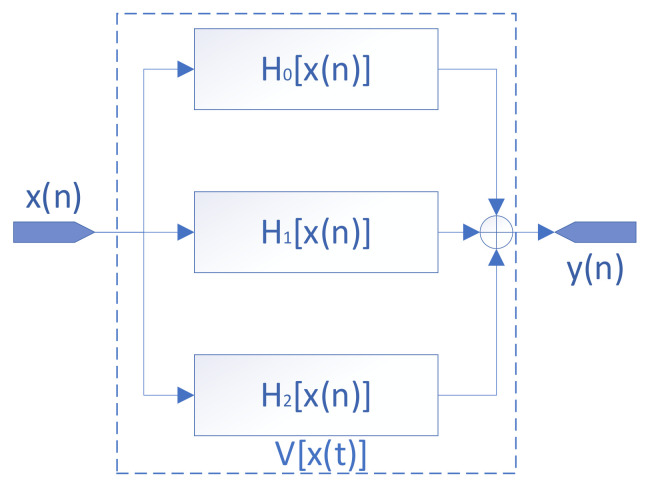
Three-order Volterra series model of a single ADC.

**Figure 3 sensors-22-02427-f003:**
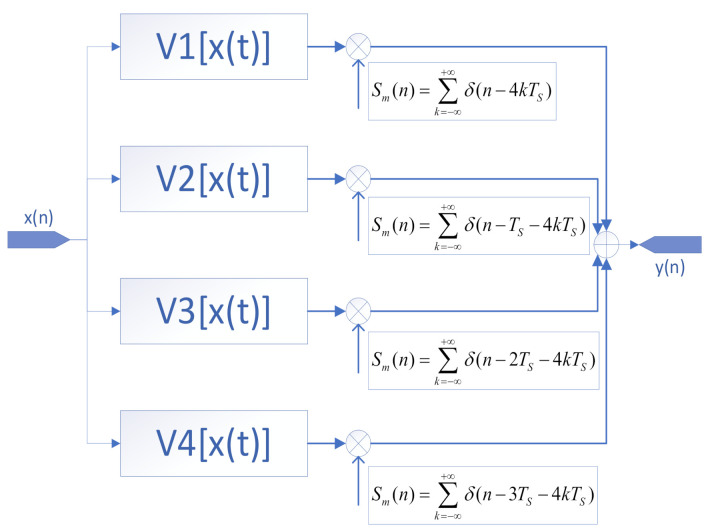
Three-order Volterra series model of the TIADC.

**Figure 4 sensors-22-02427-f004:**
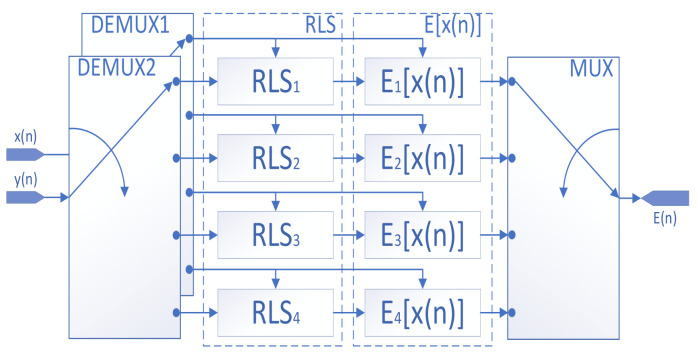
Concrete error estimation and reconstruction structure of the TIADC.

**Figure 5 sensors-22-02427-f005:**
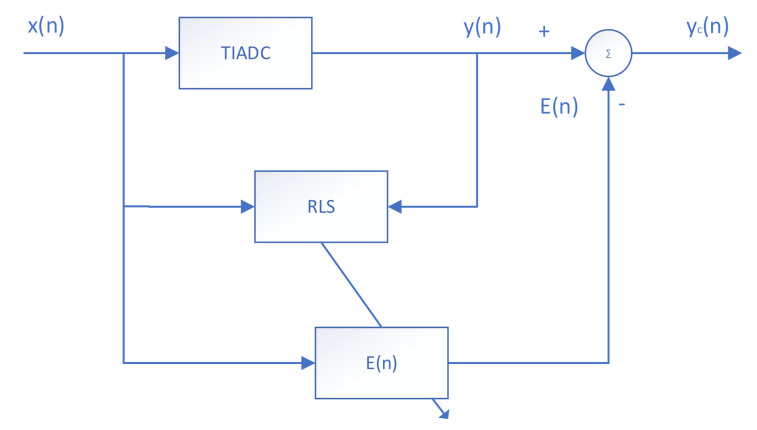
Overall error reconstruction and compensation structure of the TIADC.

**Figure 6 sensors-22-02427-f006:**
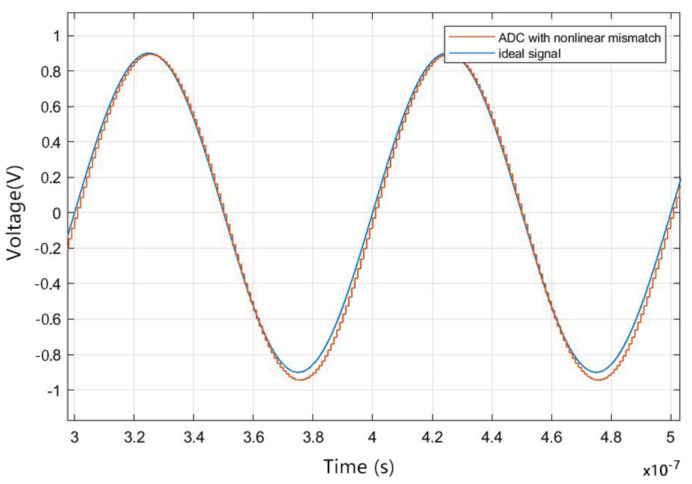
The time-domain figure of a single ADC with a nonlinear mismatch.

**Figure 7 sensors-22-02427-f007:**
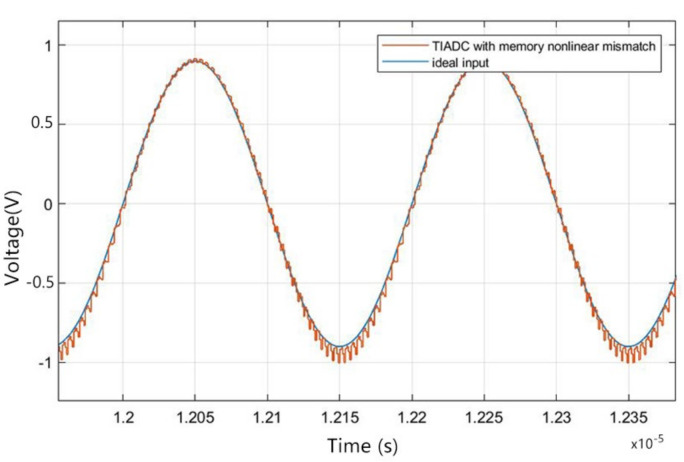
Time-domain figure of the TIADC model with memory nonlinearity mismatch.

**Figure 8 sensors-22-02427-f008:**
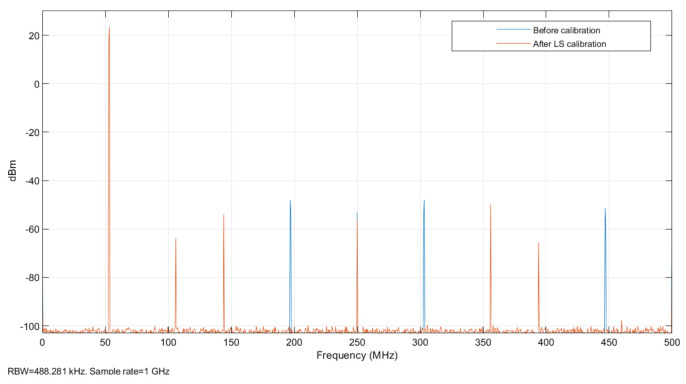
Spectrum of the TIADC model with memory nonlinearity mismatch.

**Figure 9 sensors-22-02427-f009:**
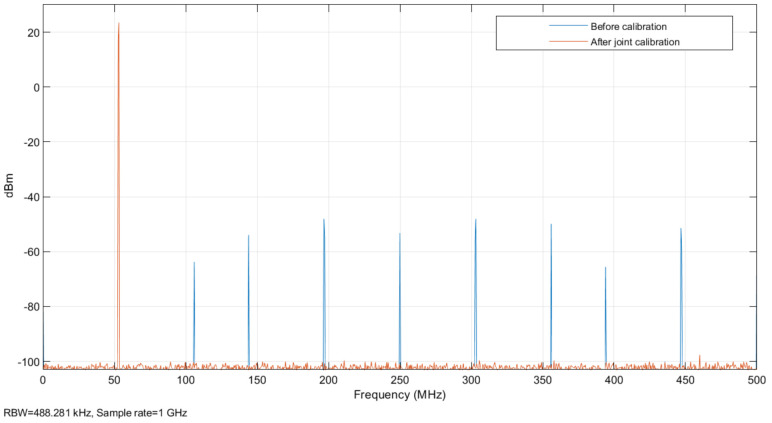
Spectrum of the TIADC model calibrated by offline LS estimation.

**Figure 10 sensors-22-02427-f010:**
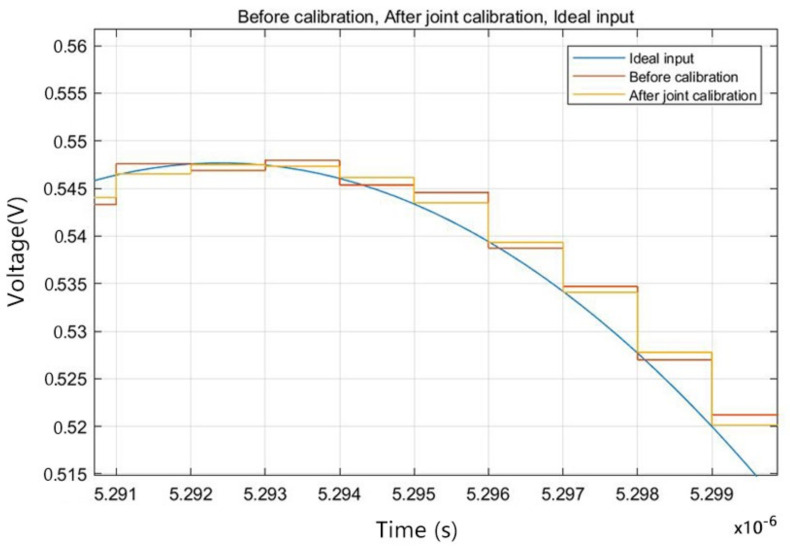
Time-domain output of the calibration system.

**Figure 11 sensors-22-02427-f011:**
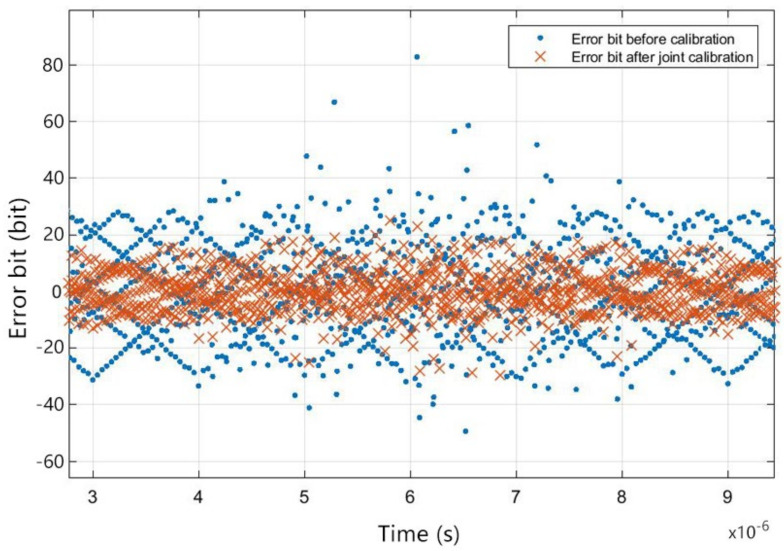
Error bit comparison before and after calibration.

**Figure 12 sensors-22-02427-f012:**
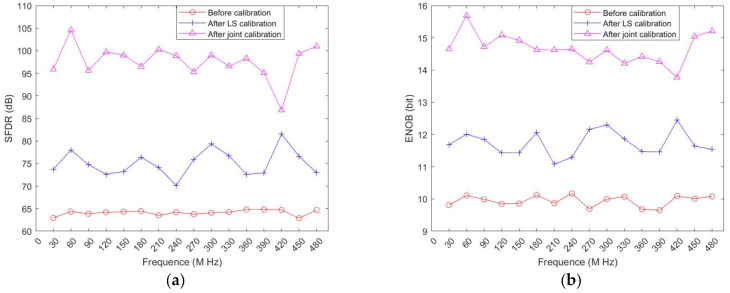
(**a**) SFDR of different stages; (**b**) ENOB of different stages.

**Table 1 sensors-22-02427-t001:** Final calibration effect comparison of the 16-bit ADC.

The Calibration Mode	SFDR (dB)	SNR (dB)	ENOB (Bit)
Before calibration	64.29	62.35	10.06
After LS calibration	76.88	74.45	12.06
After joint calibration	100.51	94.79	15.42
**Final Increment**	36.22	32.44	5.36

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
