# Peer review of "Joint Error Estimation and Calibration Method of Memory Nonlinear Mismatch for a Four-Channel 16-Bit TIADC System"

_sensors, 2022, doi:10.3390/s22072427_

Round 1

Reviewer 1 Report

The authors have investigated a joint error estimation method to address the memory nonlinear mismatch problem of the time-interleaved ADC system. Analyses and discussions are interesting and helpful for the research of the ADC components in communication systems. The paper can be published in Sensors, provided following issues can be addressed.

  1. Some abbreviations should be clarified when they appear for the first time.
  2. Add a table to describe the physical parameters of the considered setup.
  3. Add some discussion regarding the impact of channel distortions on the performance of the proposed method.
  4. Adding some discussion regarding the proposed scheme in the system with multiple-order modulation formats, such as QPSK, 16-QAM and 64-QAM. Discuss if there exists any modulation format dependency in the proposed technique.

See e.g.

Á. F. Bocco, Background compensation of static TI-ADC nonlinearities in coherent optical receivers, IEEE Argentine Conference on Electronics, 2021.

T Xu et al., Information rates in Kerr nonlinearity limited optical fiber communication systems, Optics Express, 2021.

Author Response

Dear reviewer:

I am very grateful to your comments for the manuscript. According your advice, we amended the relevant part in manuscript. The two questions you raised were answered below.

1)Thank you for the reminder, now we clarified abbreviations in the body of our manuscript now.

2)At your suggestion, we have modified the content of our table to make it easier for readers to understand the comparison between the method proposed in this manuscript and the traditional method.

3)We add some discussion regarding the impact of channel distortions on the performance of the proposed method, and then we repeated the experiment and modified Figure 7 and add part of the discussion about the impact of channel distortions.

4)The manuscript we submitted mainly studied the nonlinear influence in the quantization process of ADC, which did not involve the influence of modulation. However, according to the article (T Xu et al., Information rates in Kerr nonlinearity limited optical fiber communication systems, Optics Express, 2021.) you provided, we thought that some of the content could be used for reference by us, and we quoted some of the content in our discussion.

Thank you very much.
Sincerely,
The author of the manuscript

Reviewer 2 Report

I have two comments/questions on the content:

  1. What would be the time consumption comparison between no Calibration/LS calibration/joint calibration? And what would be the time response in terms of different volterra series model?
  2. Please keep the format clean and have all formulas in same font, and not to extend the content width.  

Author Response

Dear reviewer:

I am very grateful to your comments for the manuscript. According your advice, we amended the relevant part in manuscript. The two questions you raised were answered below.

1)In this manuscript, the comparison of the TIADC dynamic nonlinear calibration by different method are under the same time, although the time consumption between no calibration/LS calibration/joint calibration also different, but we think the difference mainly appeared on the amount of calculation, the traditional LS method in the real-time background calibration of dynamic error needs large amount of calculation, at the same time, traditional method needs to store more date which needs more memory space. In order to achieve the purpose of background calibration, more hardware resources are consumed, so the time consumption is not analyzed in the original manuscript, only the time complexity of the computation is analyzed, now we have made some changes to this part of the discussion.

2)Under your reminding, we found that there were indeed many formulas with different formats, and we corrected them all. As for the inconsistent content widths you mentioned, we found that the formula was too long, so we scaled it now.

Thank you very much.

Sincerely,

The author of the manuscript

Reviewer 3 Report

The manuscript (sensors-1632647), Joint Error Estimation and Calibration Method of Memory Nonlinear Mismatch for a Four-channel 16 Bit TIADC System, shows an interesting result of the method for a Four-channel 16 Bit TIADC System. Authors presented quite comprehensive analysis and detailed methods in this manuscript. Some comments would like to provide here for authors' reference to further improve their work - shown as following, 

  1. For the fitting results in this work, authors should provide the R_square values (coefficient of determination) for this referee and potential readers reference.
  2. Please provide a simple benchmark Table to summarize and compare the current work with previous work (e.g., what key performance or metrics authors think that would be better than previous work). This would be quite helpful for this referee and potential readers to get further insight of this research work contribution and impact in this work as compared to other works. 

Due to the above comments, this referee would like to put the manuscript status as "Minor Revision" in the current phase. 

Author Response

Dear reviewer:

I am very grateful to your comments for the manuscript. According with your advice, we amended the relevant part in manuscript. The questions you raised were answered below.

1)RLS method with forgetting factor was not adopted in this paper, so we did not provide coefficient of determination in this paper. However, under your reminding, we added Volterra series kernel and other key parameters used in building the model to the manuscript for potential readers' reference.

2)At your suggestion, we have modified the content of our table to make it easier for readers to understand the comparison between the method proposed in this manuscript and the traditional method 

Thank you very much.

Sincerely,

The author of the manuscript

Reviewer 4 Report

This paper discusses the use of Least Squares (LS) and Recursive Least Squares (RLS) for nonlinear error estimation in Time-Interleaved Analog-to-Digital Converters (TIADC).  Error estimation is used to predict the error out of the TIDAC and compensate for it, thus increasing the accuracy of its output. The paper also demonstrated that the SNR and the effective number of bits (ENOB) for the TIADC are improved using this approach.

This paper is novel and its results are significant and worthy of publishing.

I have a couple of minor revisions that I would like the request the authors to incorporate before publication:

  1. ADC, TIADC, LS, RLS are all terms defined in the abstract but not in the body of the paper. Please make sure to introduce all these abbreviations in the body of the paper and the abstract.
  2. Figures 6 and 7 look identical to me. You may want to draw these two plots in the same graph and zoom-in to show the difference between the two plots (with and without nonlinear mismatch).

Author Response

Dear reviewer:

I am very grateful to your comments for the manuscript. According your advice, we amended the relevant part in manuscript. The questions you raised were answered below.

1)Thank you for the reminder, now we clarified abbreviations in the body of our manuscript now.

2)In figure 7,we forgot add channel distortions to the TIADC system model we use, which makes it looks the same as figure 6,now, now we have corrected it.

In addition, we have modified some of the little problems you raised, such as moderate English, modified the table etc.

Thank you very much.

Sincerely,

The author of the manuscript